# High-Pressure Synthesis, Synchrotron Single-Crystal XRD and Raman Spectroscopy of Synthetic K–Ba Minerals of Magnetoplumbite, Crichtonite and Hollandite Group Indicatory of Mantle Metasomatism

Valentina Butvina [1,*] , Anna Spivak [1] , Tatiana Setkova [1] and Oleg Safonov [1,2,3]

[1] D.S. Korzhinskii Institute of Experimental Mineralogy, Russian Academy of Sciences, Academician Osipyan Str., 4, Chernogolovka, Moscow 142432, Russia
[2] Geological Department, Lomonosov Moscow State University, Vorob'evy Gory, Moscow 119899, Russia
[3] Department of Geology, University of Johannesburg, Auckland Park, Johannesburg 2006, South Africa
* Correspondence: butvina@iem.ac.ru

**Abstract:** The paper summarizes the results of an experimental study of the formation of K–Ba high-Ti (and Cr) oxides synthesized in the chromite–rutile/ilmenite–$K_2CO_3$/$BaCO_3$–$H_2O$–$CO_2$ systems at 1.8–5.0 GPa. Experiments confirm the conclusion that the formation of K–Ba high-Ti oxides characterizes the most advanced or repeated metasomatic stages in upper mantle peridotites, which lead first to the formation of simple Ti oxides and then to the formation of K–Ba high-Ti and Cr oxides. Relations between the oxides is a function of the activity of the K and Ba components in the fluid. The appearance of priderite corresponds to the highest activity of K in the mineral-forming media. Redledgeite is formed only in the Fe-poor chromite–rutile–$H_2O$–$CO_2$–$BaCO_3$ system, and, in the system with ilmenite, minerals of the magnetoplumbite group preferably crystallize. A direct dependence of the Cr content in oxides on pressure is revealed. Raman spectra of K–Ba high-titanium oxides are presented. The structure of a potassium compound of a magnetoplumbite group with the chemical formula $K_{0.90}Ti_{5.16}Cr_{2.94}Fe_{2.54}Mg_{0.87}Al_{0.22}Mn_{0.30}O_{19}$ is studied by single-crystal X-ray diffraction using a synchrotron radiation. The obtained data can be used to specify the nomenclature of the magnetoplumbite mineral group.

**Keywords:** experiment at high PT parameters; magnetoplumbite; crichtonite and hollandite groups; Raman spectroscopy; crystal structure; yimengite

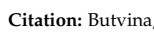



## 1. Introduction

Metasomatic Ti-rich oxides in mantle xenoliths from kimberlites and other deep rocks are widespread in all the shells of the Earth. In the upper mantle, oxides cannot compete with silicates in terms of volume and occupy a highly subordinate position; however, wide variations in chemical composition and types of association determine the exceptional importance of oxides as petrogenetic indicators. Oxides are often very sensitive to temperature, pressure, oxygen fugacity, the composition of the rock or the formation environment as a whole. Therefore, oxides can register episodes of partial melting, the degree of melting and episodes of subsequent metasomatic processes [1]. In the 1970s, innovative work [2] led to a purposeful study of the processes of mantle metasomatism. Transformations of mantle rocks under the influence of external fluids and melts, regardless of their origin and composition, are the essence of the process, which, in the 1970s, was called "mantle metasomatism" [2–6]. In the 1980s, researchers identified two main types of mantle metasomatism: (1) "modal" metasomatism, characterized by the formation of new minerals [7], and (2) "cryptic" metasomatism, expressed only in a change in the composition of primary minerals without the formation of new phases [8]. Subsequently, the third type

of mantle metasomatism was characterized, i.e., "stealth" metasomatism [9], which results in the "refertilization" of depleted harzburgites and dunites through the formation of new garnets and clinopyroxenes in them as a result of the gain of "basalt" components (Ca, Na, Al, Fe, Ti) and various trace elements with fluids [10].

Usually, the activities of $H_2O$ and/or $CO_2$ are considered as the most important factors of mantle metasomatism [9,10]. However, there is no doubt that other components can also exhibit perfectly mobile properties in this process. They are K and Na, the participation of which is an integral characteristic of "modal" mantle metasomatism [7,9]. Natural data show a wide range of reactions controlled by K activities in metasomatized upper mantle peridotites. Transformations controlled by K activities in peridotites begin with the decomposition of Al-rich minerals (garnet and spinel) to form phlogopite and amphibole and, subsequently, other alkali-bearing phases.

Different, newly formed phases characterize different degrees of metasomatic transformations. Phlogopite is considered the most common mineral indicator of the modal mantle metasomatism. The formation of other K phases in upper mantle peridotites usually corresponds to more intense metasomatic modifications. This is K-feldspar [11], the formation of which, in addition to high-K activity, indicates low water activity. Metasomatized xenoliths contain K–Ba-bearing high-Ti oxides with a high chromium content, such as minerals of the mathiasite–lindsleyite, hawthorneite–yimengite and redledgeite–priderite groups. They are usually associated with potassium richterite, phlogopite and Al-poor clinopyroxene, in absence of garnet, while coexisting spinel is characterized by high Mg and Cr contents.

Spinel, ilmenite and Cr-rich rutile are typical associating minerals and play an important role, for example, as seeds for crystallization or as reaction products; ilmenite can also be directly metasomatic [1]. The formation of these minerals characterizes the highest degrees of metasomatic modifications under conditions of high-K activity [10]. All these conclusions are based on the study of natural parageneses, as well as model thermodynamic calculations [10]. Experimental data on reactions of mantle metasomatism of peridotites are scarce.

Experimental data on the stability of K–Ba high-Ti oxides are presented by few works on their synthesis from mixtures of simple oxides and only limit the range of P–T conditions for their possible formation [12,13]. They do not reproduce the actual reactions of the formation of these minerals in mantle assemblages via the interaction of alkali-rich fluids/melts with Cr and Ti-bearing peridotite minerals. Our previous experiments [14–16] showed the fundamental possibility of the formation of K–Ba high-Ti oxides as a result of reactions of chromite with aqueous–carbonic potassium fluids and confirmed the wide range of coexistence of various oxides with respect to the composition of the fluid.

Thus, the indicator minerals of the late stages of mantle metasomatism are high-Ti and Cr oxides enriched in K and Ba (in some cases Na and Ca) and high-charge (HFSE), light rare-earth (LREE) elements, U and Th. These are minerals belonging to the priderite group of the hollandite supergroup [17]—solid solutions of $K(Ti_7Cr)O_{16}$ (K–Cr priderite)—$K(Ti_7Fe^{3+})O_{16}$ (priderite)—$Ba(Ti_6Cr)O_{16}$ (redledgeite)—$Ba(Ti_6Fe^{3+})O_{16}$ (Ba–priderite); the crichtonite group—solid solution $K(Ti, Cr, Fe, \dots)_{21}O_{38}$–$Ba(Ti, Cr, Fe, \dots)_{21}O_{38}$ (mathiasite–lindsleyite); and the magnetoplumbite group—solid solution $K(Ti, Cr, Fe, \dots)_{12}O_{19}$–$Ba(Ti, Cr, Fe, \dots)_{12}O_{19}$ (yimengite–hawthorneite). This paper presents the results of the synthesis of K–Cr and Ba–Cr end members of solid solutions of titanates (redledgeite, priderite, hawthorneite, yimengite and lindsleyite and mathiasite) as a result of the interaction of spinel–ilmenite (or rutile) associations with the fluids $H_2O$–$CO_2$-$K_2CO_3$/$BaCO_3$ at 1.8–5.0 GPa and 1000–1200 °C.

In the paper, we summarize our recent results of an experimental study on the formation of K–Ba high-Ti and Cr oxides in the chromite–rutile/ilmenite–$K_2CO_3$/$BaCO_3$–$H_2O$–$CO_2$ systems at 1.8–5.0 GPa. We report new data on Raman spectra of some of these compounds, as well as new data on the structure of a potassium compound of a magnetoplumbite group. The data are significant for the interpretation of mineral assemblages forming via metasomatism in the upper mantle.

## 2. Materials and Methods

### 2.1. Experimental Methods

The experiments were carried out in the spinel–ilmenite/rutile–$H_2O$–$CO_2$–$K_2CO_3$/ $BaCO_3$ systems. At 5 GPa and 1200 °C, the experiments were performed using a toroidal "anvil-with-hole" high-pressure apparatus NL-13T. At 3.5 GPa and 1200 °C, the experiments were performed using a "anvil-with-hole" high-pressure apparatus NL-40, and, at 1.8–2.0 GPa, assemblies were performed using a "piston cylinder" PC-40. All experiments simulated the P–T conditions of the upper mantle in the physicochemical experiment. As a starting material, there was used a mixture of chromite $(Mg_{0.49–0.54}Fe_{0.50–0.54}Mn_{0.01–0.02}Zn_{0.01–0.02})$ $(Al_{0.17–0.20}Cr_{1.55–1.61}Fe_{0.10–0.22}Ti_{0.03–0.07})O_4$ from xenolith of garnet peridotite from the kimberlite pipe (Pionerskaya, Arkhangelsk region, Russia), ilmenite $Fe_{0.98}Mg_{0.01}Mn_{0.06}Ti_{0.93}Al_{0.01}$ $Nb_{0.01}O_3$, represented by a xenocrystal from the kimberlite of the Udachnaya pipe (Yakutia, Russia) and a synthetic $TiO_2$ powder. The fluid component was prepared from a mixture of synthetic $K_2CO_3$ or $BaCO_3$ with oxalic acid. As for oxygen fugacity, following previous conclusions [14,15], the $Fe^{3+}/Fe^{2+}$ ratios in chromite in the experimental products corresponded to the values of $\Delta logfO_2$ by 1.1–1.6 logarithmic units below the FMQ buffer. Experimental conditions are given in Tables S1 and S2.

### 2.2. Analytical Methods

#### 2.2.1. Microprobe Analyses

Each run sample was embedded in epoxy and polished. After preliminary examination in reflected light on a Nikon polarization microscope, the microscopic features of run products and phase composition microprobe analyses of minerals were carried out using the CamScan MV2300 electron microscope (VEGA TS 5130MM) equipped with the EDS INCA Energy 350 electron microscope and the Tescan VEGA-II XMU microscope equipped with the EDS INCA Energy 450 and WDS Oxford INCA Wave 700 microscope at the Korzhinsky Institute of Experimental Mineralogy, Chernogolovka, Moscow region, Russia. The analysis was carried out at 20 kV, accelerating the voltage with a current beam up to 400 rA, a point size of 115–150 nm and an "excitation" zone with a diameter of 3–4 nm. The counting time was 100 s for all elements. The ZAF matrix correction was applied.

#### 2.2.2. Raman Spectroscopy

To obtain the Raman spectra, the Senterra (Bruker) microscope/spectrometer equipped with DPSS laser 532 nm and also the Renishaw RM1000 microscope/spectrometer equipped with the diode-pumped modular laser 532 nm (for barium oxides) were used. The typical parameters of the experiment were as follows: output power 20 mW, slit $50 \times 100$ mm, accumulation time 200 s (for potassium oxides) and laser output power 22 mW, slit 50 mm, accumulation time 100 s (for barium oxides). The baselining procedure was applied to the spectra. The alignment of the spectrometer was checked before being run by taking spectra on high-purity monocrystalline Si. The measured spectra were processed using Fityk 1.3.1. software (https://fityk.nieto.pl (accessed on 01 February 2023)).

#### 2.2.3. X-ray Diffraction Analysis

The crystal structure of the synthetic potassium phase with magnetoplumbite stoichiometry was solved on a single crystal based on the results of a low-temperature (T = 90 K) X-ray diffraction experiment performed at the Belok/RSA beamline ($\lambda = 0.74539$ Å) of the Kurchatov Synchrotron Radiation Source (National Research Centre "Kurchatov Institute", Moscow) [18]. The diffraction dataset was obtained by $\varphi$ scanning of a reciprocal space region in the range of angles 0°–250° with a step of 1°. The 3D set of intensities F(hkl) was integrated using the CrysAlisPro software package [19]; the spherical absorption correction was performed using the SCALE3 ABSPACK algorithm [20].

## 3. Results

### 3.1. Synthesis of Ba High-Ti Oxides

In the chromite–rutile + Ba fluid system, redledgeite (*Red*) and lindsleyite (*Ldy*) were formed. Redledgeite is a chromic analogue of Ba priderite (*Prd*) [16]. As a result of the reactions reproduced in a similar K system [15], the following phases were formed: lindsleyite, redledgeite, rutile and modified chromite at 1.8–5.0 GPa and 1000–1200 °C (Table S1). Lindsleyite formed small, xenomorphic grains (from 1:2 to 1:4) up to 10 μm in size, found mainly as inclusions in chromite (*Chr*), rutile (*Rt*) and redledgeite. Representative phase analyses are presented in Figure 1a,b and in Table S1. Redledgeite formed individual xenomorphic (Figure 1a, run Ba–Ti) and idiomorphic grains anhedral and suboctahedral in form (Figure 1b, run Ba-1.8) ranging in size from a few to 10–30 μm (with a maximum of up to 100 μm), also found as inclusions in modified chromite. Such forms of occurrence of redledgeite are known in natural associations [21]. At 1.8 GPa, 1000 °C, the following phases were formed in this system: redledgeite and hawthorneite (Figure 1b, Table S1, run Ba-1.8). Hawthorneite (*Hwt*) (Figure 1c, Table S1, run 3Ba-1) was found in the form of individual angular grains up to 80–100 μm in size, isometric shape. Lindsleyite with a size of less than 20 μm formed lighter oval or isometric inclusions in hawthorneite. Hawthorneite and lindsleyite were formed in the chromite–ilmenite + Ba fluid system at 3.5–5.0 GPa and 1200 °C. The coexisting phases were: ilmenite, rutile and recrystallized chromite. Hawthorneite crystallized as euhedral octahedral and subhedral grains up to 30–50 μm in size (Figure 1c). Lindsleyite coexisted with Ba–mica, ferrokinositalite [22] (wt %, $K_2O$, 0.62; BaO, 22.55; MgO, 8.71; $Al_2O_3$, 16.87; FeO + $Fe_2O_3$, 12.54; $SiO_2$, 23.62; $Cr_2O_3$, 3.55; $H_2O$, 3.05).

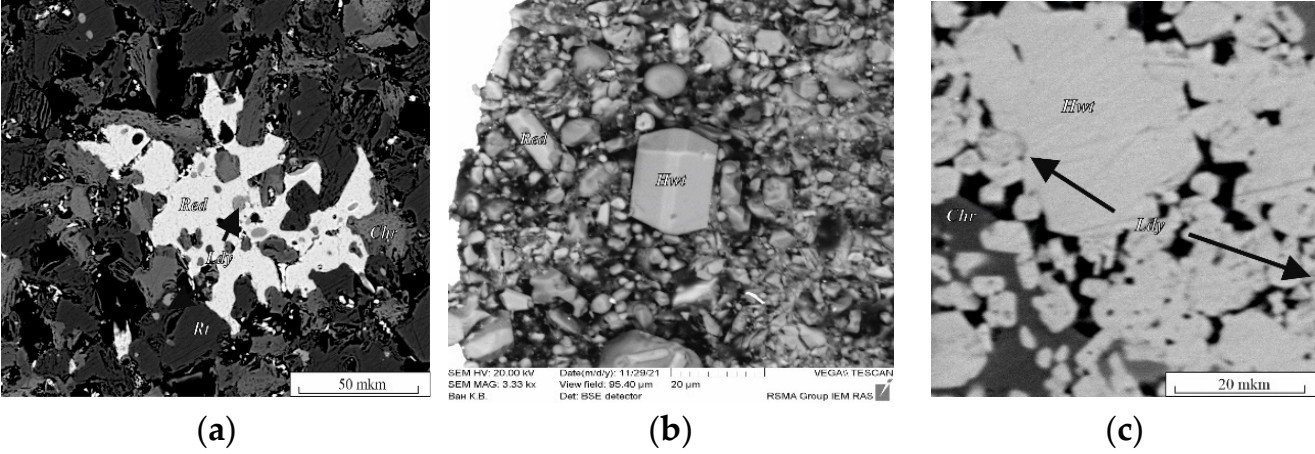

**(a)** **(b)** **(c)**

**Figure 1.** BSE images of run products in the systems: (**a**) chromite–rutile with $H_2O$–$CO_2$–$BaCO_3$ fluid; (**b**) synthetic crystals of hawthorneite and redledgeite; (**c**) chromite–ilmenite with $H_2O$–$CO_2$–$BaCO_3$ fluid (Table S1). Chr, chromite; Hwt, hawthorneite; Ldy, lindsleyite; Red, redledgeite; Rt, rutile.

The Ti–Fe–Cr diagram shows the dependence of the Cr content in Ba oxides (lindsleyite, redledgeite and hawthorneite) on the pressure in the system (Figure 2). The composition varied according to the scheme Ti ↔ Fe + Cr. There was no unambiguous dependence for K minerals of the hollandite, magnetoplumbite and crichtonite groups [15]. With increasing pressure, all Ba oxides tended to be enriched in Cr (Figure 2).

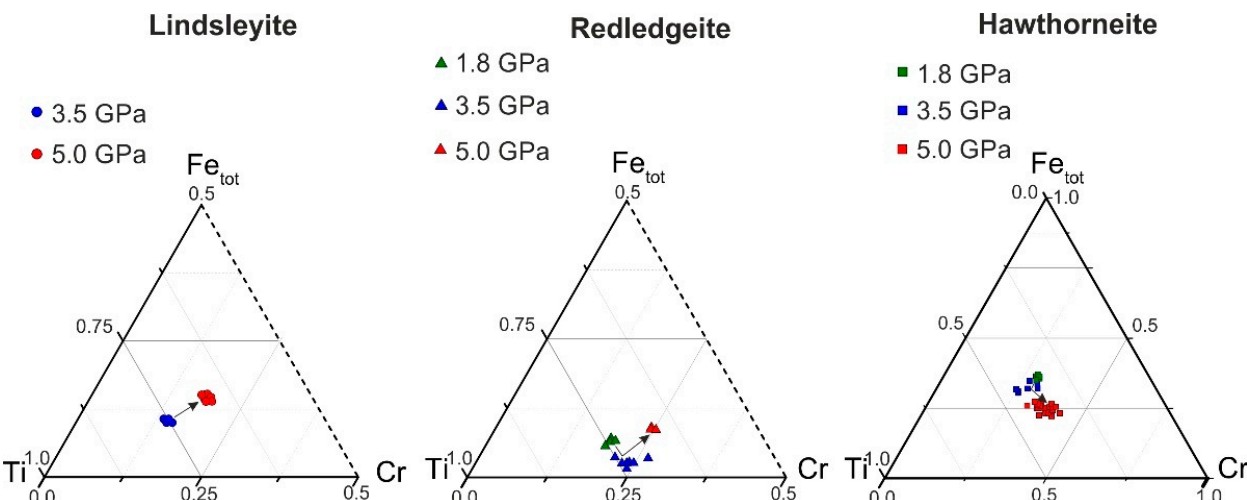

**Figure 2.** Ti–Fe–Cr diagram (apfu). The arrows indicate the displacement of points with increasing pressure.

　　　At 7–15 GPa in [9], a weakly expressed, negative, proportional dependence of the Ti content on pressure was found for experimentally obtained HAYI minerals and a negative, proportional dependence between the Ti + Mg and Cr contents was found for LIMA minerals, which also indirectly indicates a direct dependence of the Cr content in K–Ba chromium-containing oxides on pressure. For more detailed estimates of thermodynamic parameters, it is necessary to further study the isomorphic relationships of K–Ba titanates of the hollandite group, HAYI and LIMA. As a result of experimental study of the reaction of chromite and rutile, as well as chromite and ilmenite with Ba water–carbonate fluid (melt), pairs of phases of oxide (similar parageneses in K oxides [15,16]) (redledgeite, hawthorneite and hawthorneite and lindsleyite), mineral indicators of mantle metasomatism, were obtained, which directly confirm the possibility of the formation of titanium oxides as a result of mantle metasomatism of upper mantle peridotites under conditions of the highest barium activity (a possible source of barium is being discussed).

*3.2. Synthesis of K High-Titanium Oxides*

　　　The spinel–rutile + K fluid system. In association with chromite, only K–Cr priderite (*Prd*) appeared in the products of experiments with a mixture of chromite + $TiO_2$ (1:1) with a fluid (4:1 by weight) (Figure 3a, Table S2). Priderite formed idiomorphic grains up to 30–40 μm in size.

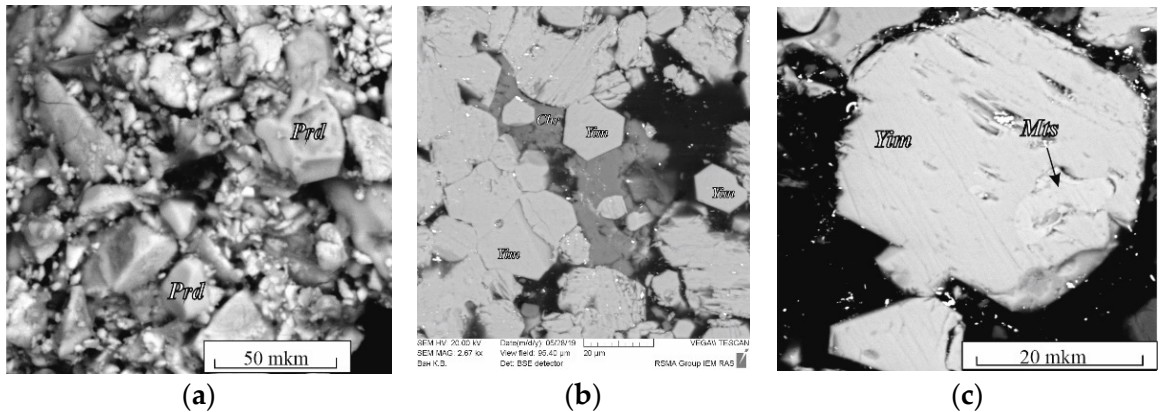

(**a**)　　　　　　　　　　　　　(**b**)　　　　　　　　　　　　　(**c**)

**Figure 3.** BSE images of run products in the systems: (**a**) chromite–rutile with $H_2O$–$CO_2$–$K_2CO_3$ fluid; (**b**) synthetic crystals of yimengite, run B1-2; (**c**) chromite–ilmenite with $H_2O$–$CO_2$–$K_2CO_3$ fluid (Table S2). Prd, priderite; Yim, yimengite; Mts, mathiasite.

In a ratio of 9:1 by weight as a result of the experiments, the joint crystallization of mathiasite and K–Cr priderite occurred. Mathiasite (*Mts*) formed subidiomorphic elongated prismatic grains (1:5) up to 200 μm in size.

The spinel–ilmenite + K fluid system. In the products of experiments involving ilmenite (experiments B1 and B2 in Table S2), along with priderite, yimengite (*Yim*) was identified, which is also associated with chromite, ilmenite and a small amount of phlogopite (probably formed due to the presence of any silicate phases—inclusions in the starting ilmenite). At $K_2CO_3/(H_2O + CO_2)$ = 7/3 and 5/5, no priderite was formed, but yimengite actively crystallized as subhedral and euhedral hexagonal crystals up to 100 μm in size (Figure 3b, c; Table S2). At lower $K_2CO_3/(H_2O + CO_2)$ ratios, no titanates were formed. Phase relations in the chromite–ilmenite + K fluid system differed at 3.5 GPa and 1200 °C. No K–Cr priderite crystallized at that pressure. Active crystallization of yimengite with inactive crystallization of mathiasite was observed in this system with a fluid ratio $K_2CO_3/(H_2O + CO_2)$ = 9/1 to 5/5 (Figure 3b, runs M; Table S2). Similar to what occurred at 5 GPa, at 3.5 GPa the crystallization ability of yimengite was at a maximum at weight proportions $K_2CO_3/(H_2O + CO_2)$ = 9/1 and 7/3 (Figure 3b). The content of this phase and the size of its crystals indicate that a pressure decrease is favorable for yimengite crystallization. As at 5 GPa, no K oxides were formed at $K_2CO_3/(H_2O + CO_2)$ lower than 5/5 at 3.5 GPa. Thus, depending on the $K_2CO_3/(H_2O + CO_2)$ ratio in the chromite–ilmenite–$K_2CO_3$–$H_2O$–$CO_2$ system, yimengite in the experimental products coexisted with potassium oxides of other structural types. At the highest ratio of $K_2CO_3/(H_2O + CO_2)$, it was accompanied by priderite, and, at lower ratios of $K_2CO_3/(H_2O + CO_2)$, priderite was not formed, and mathiasite coexisted with yimengite. The Ti–Fe–Cr diagram shows some representative compositions of K oxides (mathiasite, priderite and yimengite) [12,13,16,23–27] and our data (Figure 4). Of all the oxides, priderite is the most Fe rich. The proportions of $Fe^* = Fe^{2+} + Fe^{3+}$, Ti and Cr (p.f.u.) of this phase synthesized in systems with rutile and with ilmenite at 5 GPa were almost identical (Figure 4). A single crystal of yimengite was extracted from the sample of experiment no. B1-2 (Table S2), and its crystal structure was investigated, the results of the study of which are given in Section 3.4.

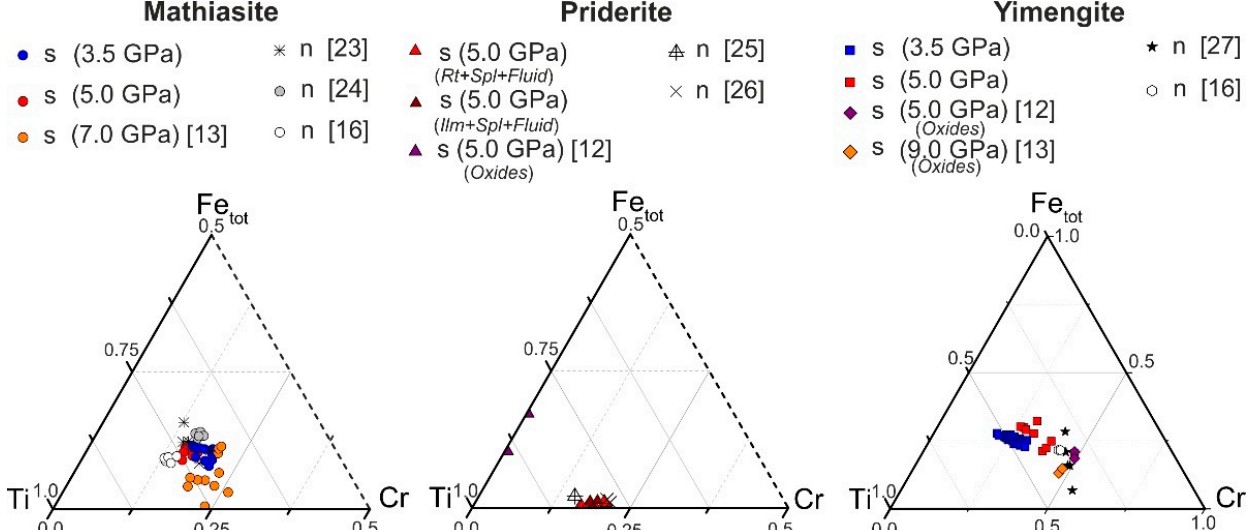

**Figure 4.** Ti–Fe–Cr diagram (apfu) illustrating compositional variations of synthetic mathiasite, priderite and yimengite. n—natural; s—synthetic.

### 3.3. Raman Spectra of Synthetic K–Ba High-Titanium Oxides

Raman spectra of the synthesized minerals were obtained in the range of 160–1000 cm$^{-1}$. Raman spectra of yimengite and hawthorneite, priderite and redledgeite and mathiasite and lindsleyite had a similar topology.

### 3.3.1. Phases of the Crichtonite Group

The Raman spectra of mathiasite and lindsleyite had several peaks (Figure 5); the strongest peak appeared at 685–687 cm$^{-1}$ and had shoulders from both sides.

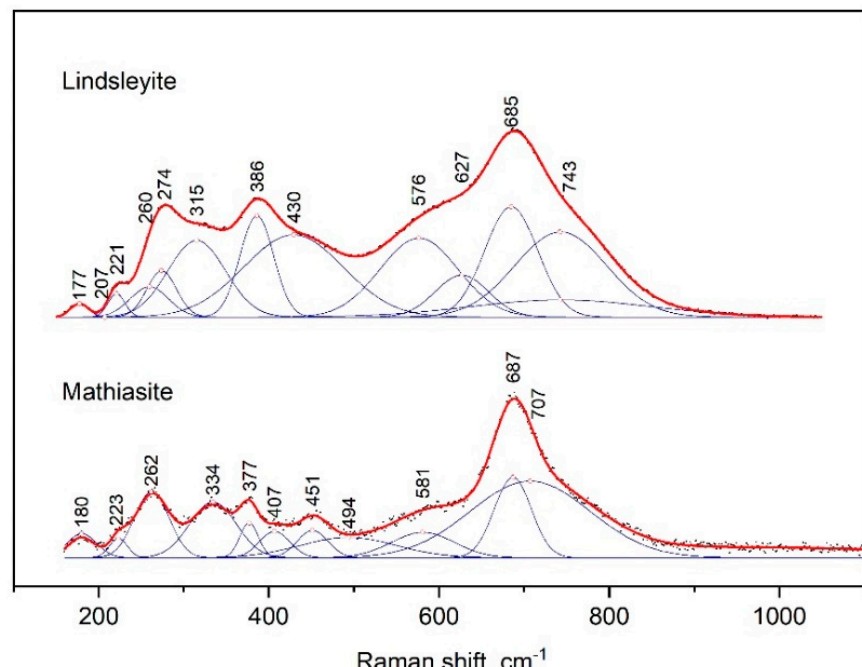

**Figure 5.** Raman spectrum of synthetic lindsleyite and mathiasite, processed using the Fityk program.

The peaks next in intensity were at 377 and 386 cm$^{-1}$, 262 and 274 cm$^{-1}$, 223 and 221 cm$^{-1}$ and 180 and 177 cm$^{-1}$ for mathiasite and lindsleyite, respectively. The spectrum of mathiasite had addition bands at 334 and 451 cm$^{-1}$.

For comparison, the Raman spectra of mathiasite and lindsleyite synthesized from oxides [13] differed in a lower Fe content. Variations in the position of all lines of the spectra shown within 5–10 cm$^{-1}$ can be explained by variations in the contents of Ca, Fe and Ti in the composition of the minerals of the crichtonite group.

### 3.3.2. Phases of the Hollandite Group

The Raman spectra of priderite and redledgeite contained two maxima at 679 and 355 and 695 and 345 cm$^{-1}$, respectively (Figure 6). The strongest peak had several shoulders from the side of large wave numbers. On the other hand, we observed shoulders at 601 and 560 cm$^{-1}$ only for redledgeite. In the case of priderite, bands were distinguished at positions 603, 563 and 501 cm$^{-1}$. The peaks at 275 and 272 cm$^{-1}$ were also present in spectra of priderite and redledgeite, respectively.

Received data were close to those for the spectra of natural K–Cr–priderite [25,26]. However, in contrast to the spectra of natural K–Cr–priderite, the main bands in the spectra of the synthetic priderite were noticeably shifted to higher wave numbers. This was probably due to the lack of Ba in the synthetic priderite.

### 3.3.3. Phases of the Magnetoplumbite Group

The Raman spectra of yimengite and hawthorneite were characterized by the strongest peak at 694 and 683 cm$^{-1}$, respectively (Figure 7). That peak represented a contribution of three successive bands from the side of large wave numbers: yimengite—763, 811 and 828 cm$^{-1}$, hawthorneite—746, 784 and 863 cm$^{-1}$; from the other side, there were additional bands at 627 and 588 cm$^{-1}$ for yimengite and 613 and 578 cm$^{-1}$ for hawthorneite. The positions of the next peaks in intensity were at 532 and 370 cm$^{-1}$ for yimengite and 528 and 348 cm$^{-1}$ for hawthorneite. For both Raman spectra in the spectral range of 200–300 cm$^{-1}$,

there was a set of weaker bands. The Raman spectra of yimengite were consistent with the spectra of the synthetic HAWYIM solid solution [13]. In comparison to the HAWYIM solid solution, the bands in the spectra of the synthesized yimengite were shifted to higher wave numbers.

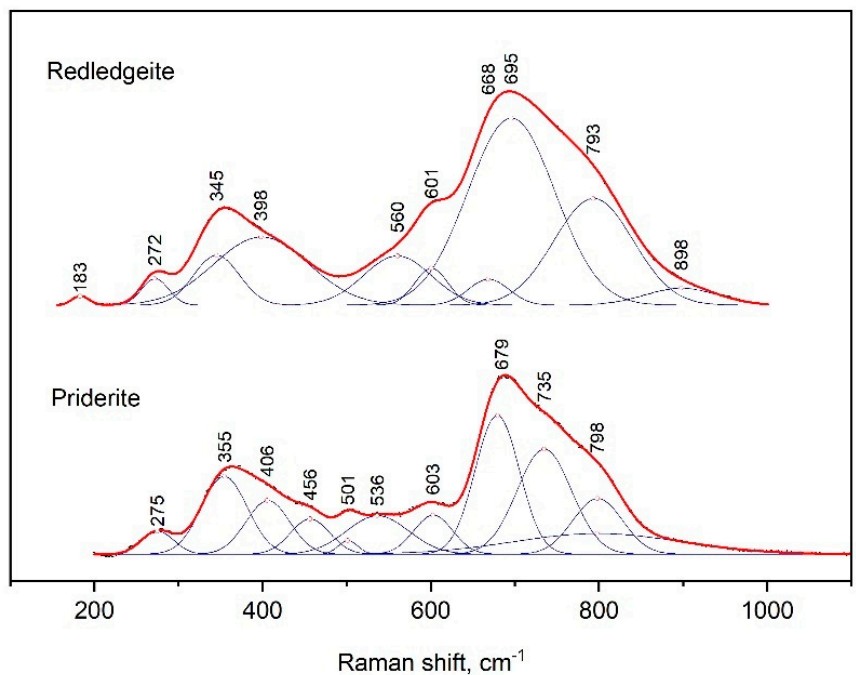

**Figure 6.** Raman spectrum of synthetic redledgeite and priderite, processed using the Fityk program.

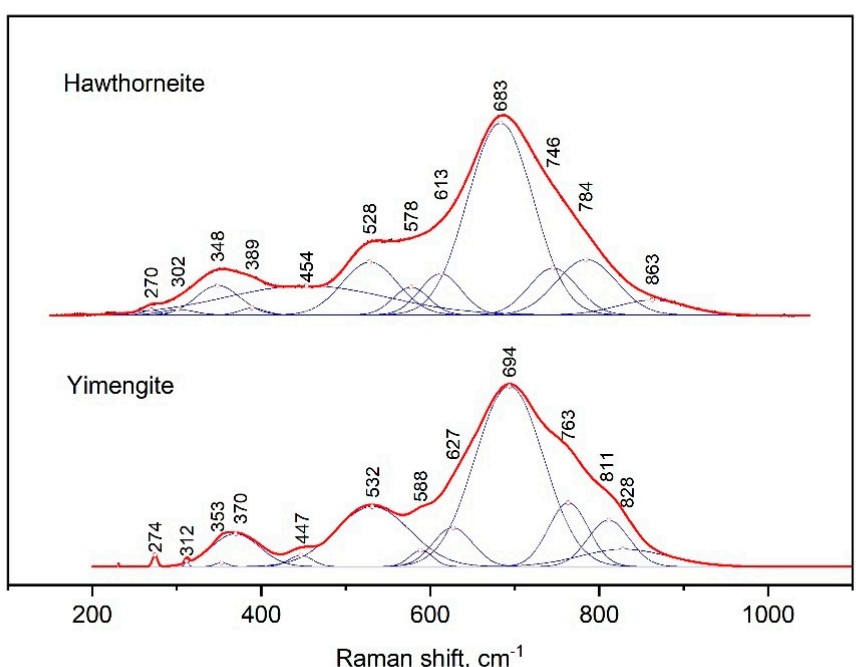

**Figure 7.** Raman spectrum of synthetic hawthorneite and yimengite, processed using the Fityk program.

### 3.4. X-ray Diffraction Analysis of Synthetic Yimengite

A single crystal (0.05 × 0.04 × 0.04 mm in size) of yimengite (Figure 8) was extracted from a polished sample. The crystal composition (MgO 4.44, $Al_2O_3$ 2.36, $K_2O$ 4.57, $TiO_2$ 41.85, $Cr_2O_3$ 23.53, MnO 2.12 and FeO + $Fe_2O_3$ 19.42 wt%) corresponded to the empirical

formula $K_{0.9}Ti_{4.87}Cr_{2.88}Mg_{1.03}Al_{0.43}Mn_{0.28}O_{19}$ (the calculation was performed based on oxygen according to the previously obtained data of the EPA). The crystal structure of yimengite was refined on a single crystal based on the results of a low-temperature (T = 90 K) X-ray diffraction experiment, performed at the Belok/RSA beamline ($\lambda$ = 0.74539 Å) of the Kurchatov Synchrotron Radiation Source (National Research Centre "Kurchatov Institute", Moscow, Russia) [18].

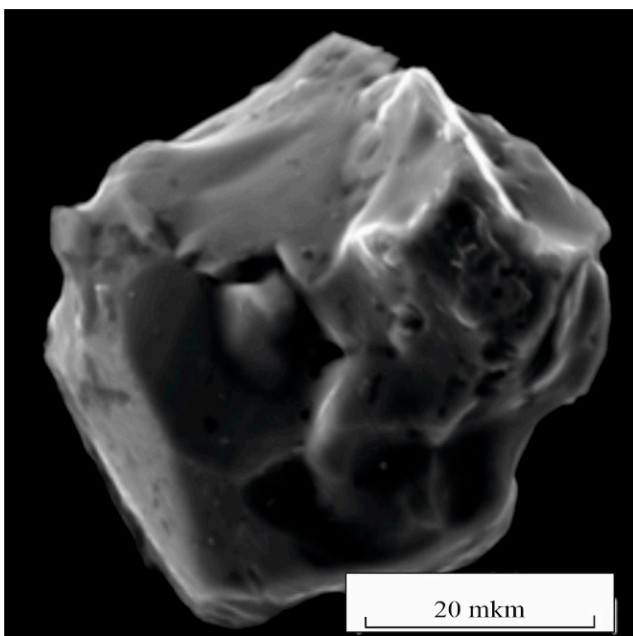

**Figure 8.** SEM image of a crystal of yimengite, selected for determining the structural characteristics.

The major crystallographic characteristics, experimental conditions and refinement parameters are listed in Table S3. The atomic coordinates and equivalent isotropic parameters for the synthetic phase structure are given in Table S4. All crystallographic data are in the cif.file in Supplementary Material. The polyhedral framework of the structure of synthesized yimengite is presented in three projections in Figure 9. The structure of this phase completely corresponded to the "ideal" magnetoplumbite structure type [28]. The atomic coordinates and atomic displacement parameters are listed in Table S5.

The alternation sequence of the structural blocks of yimengite can be presented as RSR*S* with a period c = 23.0113 (8) Å (Figure 9). The S block is a combination of two closely packed cubic layers with the structural formula. The central part of the S block contains $M_1$ octahedra (shown in green in Figure 9) and $M_3$ tetrahedra (shown in orange). The R block is a sequence of three closely packed hexagonal layers. Some of oxygen atoms in this block are replaced with cation A, as a result of which the block composition can be described by the structural formula $\{AB_6O_{11}\}^{2-}$. Trigonal $M_2$ bipyramids and $M_4$ octahedra (shown in blue and red, respectively) belong to the central part of the R block. Layers of $M_5$ octahedra (shown in gray) are located between the S and R blocks [28]. The crystallochemical formula of the synthetic yimengite (Z = 2), obtained by X-ray diffraction analysis, is in good agreement with the empirical formula. The types of atoms and occupancies of sites were determined with allowance for the balance of charges and steric factors: atomic sizes, characteristic metal–oxygen bond lengths (Table S6) and geometric parameters of the corresponding polyhedra. The values of volume, minimum, average and maximum metal–oxygen bond lengths and the Baur degree of distortion [28] for $M_i$ polyhedra are listed in Table S6. The numbers of metal atom positions are indicated in correspondence with Table S4.

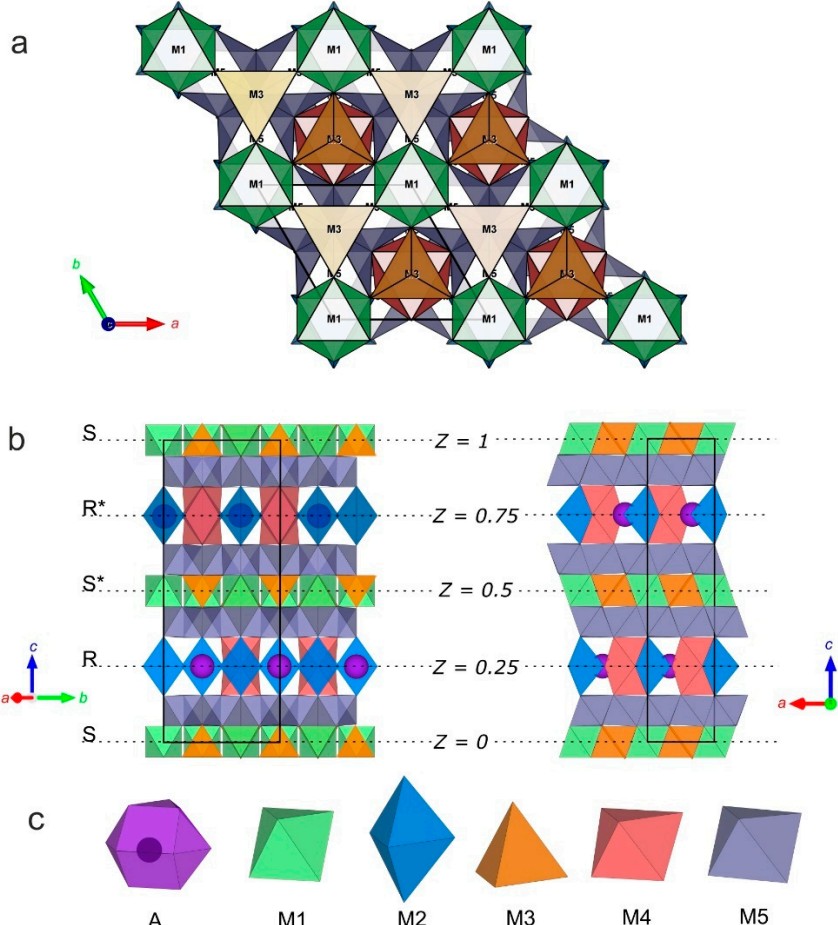

**Figure 9.** Crystal structure of synthetic yimengite: (**a**) view along the [0001] direction; (**b**) view along the [310] and [010] direction; and (**c**) types of structural polyhedra.

Thus, the synthetic potassium compound $K(Ti_5Cr_3Fe_3^{2+}Mg)O_{19}$ investigated here was isostructural to yimengite [29]. The distribution of some cations over M sites in its structure was also similar to that in yimengite [29]. The differences in the distribution of other cations can be explained by the differences in the compositions of the synthetic compound and yimengite [29]. Mn, Mg and $Fe^{2+}$ cations in both compounds were located in the $M_3$ tetrahedron (Table S7). As well as in yimengite [29], Ti in the synthetic phase was located in the $M_4$ octahedron.

Since the synthetic phase was much richer in $TiO_2$ in comparison with natural yimengite [29] (41.85 wt% vs. 29.15 wt%), and this component dominates over $Cr_2O_3$, Ti occupied not only $M_4$ but also the $M_5$ octahedron (Table S7). The decisive difference of the synthetic potassium phase from natural yimengite [29] was that all Fe in the synthetic yimengite was $Fe^{2+}$. This cation was located both in the $M_3$ tetrahedron and in the $M_5$ octahedron, similar to in hawthorneite ([30], see Table S7) and, also, haggertyite (Ba, K)($Ti_5Fe_4^{2+}$ $Fe_2^{3+}Mg)O_{19}$—a Ba–K mineral of the magnetoplumbite group [31] (Table S7). A comparison of the formulas of natural yimengite [29] and synthetic potassium phase showed that, in the absence of $Fe^{3+}$, the site of this cation was occupied by Cr and a small amount of Al, both in the $M_1$ octahedron and in the trigonal bipyramid $M_2$. In contrast to the Cr ordering in one $M_5$ site in natural yimengite, Cr was distributed over the $M_1$, $M_2$ and $M_5$ sites in the synthetic compound (Table S7).

The magnetoplumbite group minerals in mantle peridotites were presented mainly by the yimengite–hawthorneite solid solution: $K(Ti, Cr, Fe, \ldots)_{12}O_{19} - Ba(Ti, Cr, Fe, \ldots)_{12}O_{19}$. The existence of this limited solid solution is confirmed by experimental data [12]. Therefore, despite the fact that yimengite was considered (according to the published nomenclature

for the magnetoplumbite group [28]) as a mineral belonging to an individual subgroup, it is correct to nominate all minerals of the yimengite–hawthorneite solid solution as members of the same subgroup within the magnetoplumbite group.

Since the synthetic potassium compound $K(Ti_5Cr_3Fe_3^{2+}Mg)O_{19}$ investigated in this study was isostructural to yimengite [29], and its composition (with regard to the cations located in $M_{1-5}$ sites) lay within variations in the composition of natural potassium, barium and potassium–barium minerals with magnetoplumbite stoichiometry, a comparative analysis of the simplified formulas of this compound, yimengite $K(Ti_3Cr_5Fe_2^{3+}Mg_2)O_{19}$ [29], hawthorneite $Ba(Ti_3Cr_4Fe_2^{2+}Fe_2^{3+}Mg)O_{19}$ [30] and haggertyite $(Ba, K)(Ti_5Fe_4^{2+}Fe_2^{3+}Mg)O_{19}$ [31], for which structural data are absent, demonstrated some possible isomorphic substitutions, linking potassium and barium minerals of the magnetoplumbite group (Table S7). In particular, isomorphism $Ba + Fe^{2+} \leftrightarrow K + Fe^{3+}$ and $Fe^{2+} \leftrightarrow Mg$ occurred in the hawthorneite–yimengite series; the haggertyite and synthetic potassium phase were related by the isomorphism $Ba + Fe^{2+} \leftrightarrow K + Cr$ jointly with $Fe^{3+} \leftrightarrow Cr$ in the $M_2$ site.

## 4. Conclusions

Experiments on the synthesis of potassium and barium Ti- and Cr-rich oxides in the chromite–rutile/ilmenite–$K_2CO_3$/$BaCO_3$–$H_2O$–$CO_2$ systems at 1.8–5 GPa demonstrated that these minerals appear in the whole pressure range and confirmed the possibility of the coexistence of various oxides within the wide upper mantle pressure range. Although chromite is the major precursor for these minerals during reactions with K or Ba-rich aqueous–carbonic fluid, these reactions strongly require additional sources of titanium such as ilmenite and/or rutile. Availability of ilmenite or rutile determines the type of the K and Ba Ti- or Cr-rich oxide, so priderite and redlegeite form in presence of rutile, while magnetoplumbite minerals prefer ilmenite-bearing assemblages. However, since ilmenite and rutile are usually themselves produced by modal metasomatism of peridotites, the experiments confirmed that the formation of the K and Ba Ti- or Cr-rich oxides marks the most advanced stages of metasomatism in mantle peridotites. The relationships between the K and Ba Ti- or Cr-rich oxides are a function of the activity of the K and Ba components in the fluid and pressure. For example, crichtonite minerals do not form at pressures below 3 GPa. In addition, composition of these minerals with respect to the Cr, Ti and Fe cation ratio in M sites is also pressure dependent and can be calibrated further as a pressure marker.

The results on the crystal structure of the synthetic potassium magnetoplumbite compound expand the information on the distribution of A (K, Ba) and M (Ti, Cr, Fe, Mg, Al, Mn) cations over different sites in minerals with a magnetoplumbite-type structure. It was established that $Fe^{2+}$ is distributed over two nonequivalent sites $M_3$ and $M_5$. The absence of $Fe^{3+}$ in the $M_1$ and $M_2$ sites in the synthetic compound results in the distribution of Cr (and a small admixture of Al) into these sites. Cr also occupies the $M_5$ site. Relationships between the quantitative ratios of M cations and the character of their distribution over crystallographic sites with the physicochemical conditions of the formation of magnetoplumbite-type minerals are promising for evaluation the mineral genesis in the upper mantle.

**Supplementary Materials:** The following supporting information can be downloaded at: https://www.mdpi.com/article/10.3390/min13020292/s1, Table S1: Conditions and results of experiments on synthesis of redledgeite, (Red), hawthorneite (Hwt) and lindsleyite (Ldy) at 1.8, 3.5 and 5 GPa; Table S2: Conditions and results of experiments on synthesis of priderite, (Pdr), yimengite (Yim) and mathiasite (Mts) at 3.5 and 5 GPa, 1200 °C; Table S3: Main crystallographic characteristics, details of the X-ray experiment and structure refinement parameters for the synthetic yimengite; Table S4: Atomic coordinates and equivalent isotropic parameters for the synthetic yimengite; Table S5: Parameters of coordination polyhedral in the structure of synthetic yimengite; Table S6: Metal ion radii $R_M$ and metal–oxygen bond lengths l(M–O); Table S7: Cation distribution in the structures of potassium and barium minerals of the magnetoplumbite group.

**Author Contributions:** V.B. suggested a conceptual basis for the paper; V.B. and O.S. produced experimental data; A.S. and T.S. performed X-ray diffraction analysis and application of the special crystallographic program; V.B. and O.S. discussed the results and wrote the paper; V.B., A.S. and T.S. prepared the figures. All authors have read and agreed to the published version of the manuscript.

**Funding:** This research was funded by the Russian Science Foundation No. 23-27-00065, https://rscf.ru/en/project/23-27-00065/ (accessed on 01 February 2023).

**Acknowledgments:** Authors thank of V.N. Kovalev for help in crystal structure visualization. Authors also thank the three anonymous reviewers for their constructive suggestions.

**Conflicts of Interest:** The authors declare no conflict of interests.

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
