# Peer review of "High-Pressure Synthesis, Synchrotron Single-Crystal XRD and Raman Spectroscopy of Synthetic K–Ba Minerals of Magnetoplumbite, Crichtonite and Hollandite Group Indicatory of Mantle Metasomatism"

_minerals, doi:10.3390/min13020292_

Round 1
Reviewer 1 Report
Butvina et al. report the experimental investigation of synthetic minerals containing K-Ba in the families of magnetoplumbite, crichtonite, and hollandite. These minerals are partially responsible for transformations of mantle rocks under the influence of external fluids and melts. Authors have performed high pressure syntheses (up to 5 GPa), Raman spectroscopy, and single crystal X-ray diffraction (yimengite/magnetoplumbite only). Manuscript is well written. Conclusions are consistent with the experimental data and their analyses. Length of the manuscript is good. I’d recommend this work for publication in Minerals after some minor revision. In more details:
1 For the Raman measurements, what was the sample preparation used? Please include this info in the revised manuscript
2. For the single crystal X-ray analyses:
a. Was the nominal composition used as a starting point and then calculations based on cations and oxygen were utilized?
b. Was the composition measured by energy (EDS) or wavelength (WDS) dispersive spectroscopy during microprobe analyses?
c. Please provide occupancy fractions in Table S4 and submit a CIF as supplementary material, so that readers have easy access to the proposed structure
d. Were the occupancy fractions fixed based on composition or freely refined?
e. Please include error bars in the proposed stoichiometry of yimengite
3. Page 9, line 303: Please provide the standard error next to the length of the c-axis
Author Response
Butvina et al. report the experimental investigation of synthetic minerals containing K-Ba in the families of magnetoplumbite, crichtonite, and hollandite. These minerals are partially responsible for transformations of mantle rocks under the influence of external fluids and melts. Authors have performed high pressure syntheses (up to 5 GPa), Raman spectroscopy, and single crystal X-ray diffraction (yimengite/magnetoplumbite only). Manuscript is well written. Conclusions are consistent with the experimental data and their analyses. Length of the manuscript is good. I’d recommend this work for publication in Minerals after some minor revision. In more details:
- For the Raman measurements, what was the sample preparation used? Please include this info in the revised manuscript
Raman spectra were collected from crystals of 20-40 microns in size directly in the polished run samples mounted in epoxy and preliminarily examined for homogeneity using electron microscope and microprobe.
- For the single crystal X-ray analyses:
- Was the nominal composition used as a starting point and then calculations based on cations and oxygen were utilized?
the calculation was performed based on oxygen, according to the previously obtained data of the EPA
- Was the composition measured by energy (EDS) or wavelength (WDS) dispersive spectroscopy during microprobe analyses?
The composition was measured by energy (EDS) analyses. Polished samples were preliminarily coated with ~ 15 µm carbon film. The microprobe measurements were carried out at an accelerating voltage 20 kV, a focused beam current of ~ 10nA, and counting times of 60–120 s. For each sample, 10–20 analyzed points were measured and averaged.
- Please provide occupancy fractions in Table S4 and submit a CIF as supplementary material, so that readers have easy access to the proposed structure
The data in the table has been corrected. I submit CIF file as supplementary material
- Were the occupancy fractions fixed based on composition or freely refined?
the occupancy fractions were fixed based on composition
- Please include error bars in the proposed stoichiometry of yimengite
if we are talking about graphs, then the size of the marker exceeds the error.
- Page 9, line 303: Please provide the standard error next to the length of the c-axis
|
c [Å] |
23.0113(8) |

Reviewer 2 Report
The manuscript describes high-pressure synthesis and X-ray and Raman characterization of mantle oxide minerals with relation to the natural processes. The work provides, in general, a positive impression and can be considered for publication in Minerals (MDPI). My major concern is, however, the readability of the text. The names of the mineral repeatedly described in the text are rather effort-demanding upon reading by a general audience. To improve this point, I suggest using abbreviations (like, for instance, in Fig1) and introducing them just once.
Some minor points:
1. Please clarify the statement "the calculation was performed using both the cation-based and oxygen-based methods"
2. Please eliminate Cyrillic letters (Fig8, Table S3)
3. Please add information on completeness of the X-ray data set and <I/sigma>. Why only spherical absorption correction was done? It is desirable to provide cif file of the solved structure along with crystallographic data.
Author Response
The manuscript describes high-pressure synthesis and X-ray and Raman characterization of mantle oxide minerals with relation to the natural processes. The work provides, in general, a positive impression and can be considered for publication in Minerals (MDPI). My major concern is, however, the readability of the text. The names of the mineral repeatedly described in the text are rather effort-demanding upon reading by a general audience. To improve this point, I suggest using abbreviations (like, for instance, in Fig1) and introducing them just once.
We do not see any problem in reading the mineral names. As a rule, abbreviations of minerals are used on graphs and in tables. It would not be correct to use abbreviations of minerals-oxides in the text, and to use the names in silicate minerals in full
Some minor points:
1 Please clarify the statement "the calculation was performed using both the cation-based and oxygen-based methods"
the calculation was performed based on oxygen, according to the previously obtained data of the EPA
- Please eliminate Cyrillic letters (Fig8, Table S3)
Have done
3 Please add information on completeness of the X-ray data set and <I/sigma>. Why only
spherical absorption correction was done? It is desirable to provide cif file of the solved structure along with crystallographic data.
According to the technical and methodological capabilities of the device, only correction of spherical absorption was performed. I submit CIF file with crystallographic data as supplementary material
